# A rapid rate of sex-chromosome turnover and non-random transitions in true frogs

Daniel L. Jeffries [1], Guillaume Lavanchy[1], Roberto Sermier[1], Michael J. Sredl[2], Ikuo Miura[3], Amaël Borzée [4], Lisa N. Barrow[5], Daniele Canestrelli [6], Pierre-André Crochet[7], Christophe Dufresnes[1], Jinzhong Fu[8], Wen-Juan Ma[1], Constantino Macías Garcia [9], Karim Ghali[1], Alfredo G. Nicieza[10,11], Ryan P. O'Donnell[2], Nicolas Rodrigues [1], Antonio Romano[12,13], Íñigo Martínez-Solano[14], Ilona Stepanyan[15], Silvia Zumbach[16], Alan Brelsford[17] & Nicolas Perrin [1]

The canonical model of sex-chromosome evolution predicts that, as recombination is suppressed along sex chromosomes, gametologs will progressively differentiate, eventually becoming heteromorphic. However, there are numerous examples of homomorphic sex chromosomes across the tree of life. This homomorphy has been suggested to result from frequent sex-chromosome turnovers, yet we know little about which forces drive them. Here, we describe an extremely fast rate of turnover among 28 species of Ranidae. Transitions are not random, but converge on several chromosomes, potentially due to genes they harbour. Transitions also preserve the ancestral pattern of male heterogamety, in line with the 'hot-potato' model of sex-chromosome transitions, suggesting a key role for mutation-load accumulation in non-recombining genomic regions. The importance of mutation-load selection in frogs might result from the extreme heterochiasmy they exhibit, making frog sex chromosomes differentiate immediately from emergence and across their entire length.

[1] Department of Ecology and Evolution, University of Lausanne, CH-1015, Lausanne, Switzerland. [2] Arizona Game and Fish Department, 5000 W. Carefree Highway, Phoenix, AZ 85086, USA. [3] Amphibian Research Center, Hiroshima University, Higashi-Hiroshima 739-8526, Japan. [4] Division of EcoScience and Department of Life Sciences, Ewha Womans University, Seoul 03760, Republic of Korea. [5] Museum of Southwestern Biology, MSC03 2020, 1 University of New Mexico, Albuquerque, NM 87131, USA. [6] Department of Ecological and Biological Science, University of Tuscia, 01100, Viterbo, Italy. [7] CEFE, CNRS, University Montpellier, University Paul Valéry Montpellier 3, EPHE, IRD, Route de Mende, Montpellier 34293, France. [8] Department of Integrative Biology, University of Guelph, Guelph, ON N1G 2W1, Canada. [9] Instituto de Ecología, Universidad Nacional Autónoma de México, Ciudad Universitaria, Mexico City, 04500, Mexico City, Mexico. [10] Research Unit of Biodiversity, UO-CSIC-PA, 33006, Mieres, Spain. [11] Department of Biology of Organisms and Systems, Universidad de Oviedo, 33600, Mieres, Spain. [12] Consiglio Nazionale delle Ricerche, Istituto per i sistemi agricoli e forestali del mediterraneo, Via Patacca 84, I-80056, Ercolano, NA, Italy. [13] MUSE - Museo delle Scienze, Sezione di Zoologia dei Vertebrati, corso del Lavoro e della Scienza 3, 38122, Trento, Italy. [14] Department of Biodiversity and Evolutionary Biology, Museo Nacional de Ciencias Naturales, CSIC, José Gutiérrez Abascal, 2, Madrid 28006, Spain. [15] Scientific Center of Zoology and Hydroecology, National Academy of Science, Republic of Armenia, P. Sevak str. 7, 0014, Yerevan, Armenia. [16] Info Fauna - karch, UniMail, Bellevaux 51, 2000, Neuchâtel, Switzerland. [17] Evolution, Ecology, and Organismal Biology Department, University of California Riverside, Riverside, CA 92521, USA. These authors contributed equally: Daniel L. Jeffries, Guillaume Lavanchy. Correspondence and requests for materials should be addressed to D.L.J. (email: dljeffries86@gmail.com)

Sex chromosomes experience very different evolutionary forces compared to the rest of the genome, owing to their differential occurrence in males and females. A male-beneficial mutation occurring close to a newly evolved male-determining locus, for instance, is expected to spread, even if highly detrimental to females, because linkage with the sex-locus makes it more likely to be transmitted to sons than daughters. The canonical model of sex-chromosome evolution predicts that a region of suppressed recombination will then expand outwards from the sex-determining locus, possibly via inversions, thus increasing linkage with sexually antagonistic genes[1,2]. Over time, however, this loss of recombination will lead to the accumulation of potentially deleterious mutations, repetitive elements and gene loss in sex-limited chromosomes (Y or W) due to Hill–Robertson interactions[3] and Muller's Ratchet[4]. This progressive build-up of differences between gametologs (i.e. between X and Y, or between Z and W) is thought to have given rise to the highly hetero-morphic sex chromosomes prevalent in many clades, including mammals, birds, flies and some snakes[5–7].

However, more and more examples of homomorphic sex chromosomes are now being discovered, notably in the majority of poikilothermic vertebrates, i.e. in most fish, lizards and amphibians[8,9]. This homomorphy implies that gametologs have recombined relatively recently in these taxa, and one main hypothesis for how this can occur is through recurrent sex-chromosome turnovers, i.e. a switch in the chromosome pair used for sex determination[10,11]. In systems where a turnover has recently occurred, sex chromosomes will be relatively young; thus gametologs will not have had time to substantially differentiate from one another. Indeed, frequent turnovers have been docu-mented in several lineages of fish[12,13], amphibians[14–17] and reptiles[18], and are thought to have contributed to the homo-morphy of sex chromosomes in these species. However, it should be noted that turnover rates have sometimes been inferred from changes in the patterns of heterogamety[18,19], which can grossly underestimate true values. Indeed, several groups seemingly show constant heterogamety despite high turnover rates[13,14,20,21].

Despite the importance of turnovers for sex-chromosome evolution, one important question remains unanswered: why do some taxa fall into the evolutionary trap of extensive sex-chromosome differentiation and degradation, while others can avoid this by continuously swapping them? Answering this question requires knowledge of the evolutionary forces driving turnovers. A turnover can occur if the existing sex-locus is translocated to an autosome[22,23], or if a new gene acquires the sex-determining role via mutation[24,25]. The subsequent fixation of the new sex chromosome is thought to be mediated by one (or a combination) of four main evolutionary forces[26], namely (i) genetic drift[27–29]; (ii) sex-ratio selection, induced by sex biases arising, e.g., from meiotic-drive elements or endoparasites[30–32]; (iii) sexually antagonistic selection on a gene linked to the new sex determiner[24,33]; and (iv) deleterious mutation load accumulating on the non-recombining Y or W chromosome[34,35].

The predicted patterns of turnovers under each of these forces are distinct from each other. Turnovers induced by mutation-load selection are predicted to strictly preserve the heterogametic sex, because a switch (e.g. from an XY to a ZW system) would require fixation of the heavily loaded sex-limited chromosome (Y or W) as an autosome[35]. Under drift-induced turnovers, patterns of heterogamety are generally expected to be maintained 2–4 times more often, as X or Z chromosomes are more likely to be fixed than Y or W chromosomes due to their higher initial frequency (75% vs. 25%). This ratio, however, decreases along with effective population size, and can even reverse in case of extreme polygyny, where XY becomes more likely to transition towards ZW than towards XY[29].

We also expect to see changes in the heterogametic sex via turnovers induced by either sex-ratio selection or sex-antagonistic selection: the new system of heterogamety should, in both cases, be independent of that preceding the transition, and will speci-fically depend on whether the initial sex bias was towards males or females[31,32] or the new autosomal sex-antagonistic mutation is male- or female-beneficial[24,33]. The relative importance of these driving forces for turnovers is hotly debated, but by testing for the predicted patterns of heterogamety outlined above, we can gain insights into which forces are the dominant drivers of sex-chromosome turnovers in a given system.

One particularly intriguing observation from studies doc-umenting sex-chromosome turnovers is that some specific gen-ome regions are used for sex determination more often than others[36]. For instance, two species of turtles have independently recruited the same autosome for sex determination[37]; two linea-ges of medaka fishes have independently co-opted Sox3 as their sex determiner[38]; and in sticklebacks, LG12 has been linked to sex at least twice independently[39]. Several other examples exist which, together, imply that certain regions of the genome are predis-posed to recruitment for sex determination, because they harbour important genes with the potential to determine sex[38,40], or to fix sexually antagonistic alleles[39]. However, to our knowledge, no statistical support has yet been provided to support this claim, likely due to the low number of species available to test it.

In this study, we examine the frequency and patterns of sex-chromosome turnover among 28 species of true frogs (Ranidae) to test whether heterogamety is more likely to be preserved or switched during turnovers and to test whether some chromo-somes are indeed more likely to be recruited for sex determina-tion than others. This family is thought to have a very high rate of sex-chromosome turnover, and anecdotal evidence points to some chromosomes being more likely to be recruited for sex determination than others[14], making it an ideal system in which to study these questions. However, as previous work was phylo-genetically blind, it has not been possible to distinguish the pat-terns of sex-chromosome recruitment from phylogenetic correlation. Here, we use Restriction site Associated DNA sequencing (RADseq) to search for sex-determination system and the identity of the sex chromosomes in 19 frog species. We couple this new data with that from ground-truthed literature to assess patterns of turnovers across a total of 28 species in a phylogenetic framework. Our results show firstly that Ranidae exhibit an extremely fast rate of sex-chromosome turnover, and that some chromosomes are indeed significantly more likely to be recruited for sex determination than others. Secondly, we find that in all but one case, male heterogamety is conserved during turnovers, suggesting that mutation-load selection is an important driving force in this system. Finally, we show that, in stark contrast with the conventional model of sex-chromosome evolution, frog sex chromosomes rapidly diverge along almost their entire length due to the highly reduced rate of recombination in males, which further supports a role for the accumulating mutation load and might explain why Ranidae undergo such frequent sex-chromosome turnovers relative to other taxa.

## Results

**Identifying sex-linked markers and sex-determination systems.** In total, we produced RADseq libraries for 19 species of Ranidae (Supplementary Table 1) and also re-analyzed the previously published RADseq dataset from *Rana arvalis*[68]. The final number of assembled and retained RADtags per species dataset ranged from 5251 to 146,854 (Mean = 61,152, SD = 27,485), which contained between 2351 and 71,462 Single Nucleotide Poly-morphisms (SNPs) (Mean = 29,410, SD = 21,947) and the mean

locus read-depth per dataset was 13.9–36.8 (Mean 23.8, SD = 6.7).

Our screens for sex linkage yielded markers that fit the expectations for either an XY or a ZW system in all datasets, pointing to the widespread occurrence of false positives. We therefore used an in silico approach to test whether observed numbers of putative XY or ZW markers differed from random expectations, by permuting sex across samples 1000 times and running tests for sex-linked markers on each permutation (Supplementary Note 1). The resulting null distribution was then compared with the number of sex-linked markers identified under real sex assignments. Of the 20 species for which we analyzed RADseq data, 12 had sex-linked marker sets that passed these sex-permutation validation tests for at least one method (Supplementary Fig. 1, Supplementary Table 1). In several cases, however (see R. montezumae, R. arvalis, R. italica and R. kukunoris, Supplementary Fig. 1), sets of sex-linked markers supporting both XY and ZW systems passed validation in the same species. In R. montezumae, R. italica, and R. kukunoris, there were enough of such markers that their position could be found in the reference genome (see below). In all three cases, both XY- and ZW-like markers aligned to the same chromosome (Supplementary Fig. 2a). As it is biologically implausible that both XY and ZW sex-determination systems simultaneously exist on the same chromosome, we hypothesized that the ZW signal actually resulted from markers that were either X-limited (resulting in male hemizygosity) or had an allele specific to the paternal X chromosome. Such loci can produce almost exactly the same signal as ZW markers. For these four species, where sample availability allowed, we tested this hypothesis with several supplementary analyses. The results of these tests alongside independent lines of evidence, which are detailed in Supplementary Note 2 and Supplementary Fig. 2, allowed us to be confident that these species do indeed exhibit XY heterogamety.

The final number of sex-linked markers confidently identified per dataset varied greatly, ranging from 34 to 1925 (Supplementary Table 1). Only these markers were carried forward in the analyses and are shown in bold in Supplementary Table 1. Failure to find validated sex-linked marker sets in the eight remaining species (five of which, from our literature search, have been shown to be male heterogametic) could have been due to a number of factors, including small sample sizes (Supplementary Note 3, Supplementary Fig. 3c), the presence of several Y haplotypes in the male data (Supplementary Note 3, Subsample 2 in Supplementary Fig. 3b), extremely undifferentiated sex chromosomes (which can only be identified using families with sexed offspring) or non-genetic sex determination[41].

**Identifying the sex chromosome**. In order to identify the chromosome to which sex-linked RADtags belonged, we first anchored scaffolds from an existing fragmented R. temporaria genome assembly to the high-quality assembly of Xenopus tropicalis via intermediate alignment to high-density linkage maps from males of six R. temporaria families (see details in Supplementary Note 4 and Supplementary Fig. 4). Linkage maps contained a total of 15,313 SNPs and allowed us to assign 408.14 Mb (9.07%) of the R. temporaria assembly to X. tropicalis chromosomes (Supplementary Table 2). Aligning RADtags from our study species to these chromosome-anchored R. temporaria scaffolds allowed us to identify the chromosome pair to which they belonged. To control for mis-assembly, mis-alignment or any biases towards particular chromosomes in the RADseq data, we also aligned 1000 randomly chosen subsets of markers (each of the same size as the set of sex-linked markers) to the genome for each dataset. We only accepted a sex chromosome

identification if the number of sex-linked markers which aligned to that chromosome was significantly higher than the 99th percentile of the distribution of the randomly chosen marker subsets. Of the 12 species for which sex-linked markers were confidently identified, seven species passed these alignment validation tests (Supplementary Fig. 5, Supplementary Table 1). In these species, the number of sex-linked markers that aligned to the sex chromosome ranged from 16 (R. dalmatina) to 100 (R. italica), and in all cases, only one of the 13 chromosomes was identified as being sex-linked. We note that if a set of sex-linked markers met our validation conditions, this further increased our confidence in the sex linkage of these markers, as the likelihood of false positives clustering on a single chromosome (only when using correct sex assignments) is infinitesimal.

The reasons for the unsuccessful alignments in the five remaining species for which sex-linked markers were available were either a high divergence time between that species and R. temporaria (e.g. Pelophylax perezi, approx. 55 M years diverged) and subsequently a low overall alignment rate, a low number of sex-linked markers identified (e.g. R. latastei, 94 RADtags) or a combination of the two (e.g. R. pipiens, R. tarahumarae) (Supplementary Fig. 6). However, in one species, R. kukunoris (Muyu population), the reason was likely a high number of false-positive sex-linked markers in the dataset (Supplementary Figs. 1, 6).

Across the seven species for which a sex-linked chromosome could be reliably identified, three chromosomes out of the possible 13 were used as the sex chromosome, Chr01 (R. arvalis, R. italica, R. japonica, R. montezumae), Chr03 (R. iberica) and Chr05 (R. dalmatina, R. kukunoris) (Fig. 1, Supplementary Fig. 5). The results from the literature search, confirmed by mapping of enzyme or microsatellite markers to the X. tropicalis genome, provided sex-chromosome identities for 12 additional species and, importantly, also corroborated the finding that the same three chromosomes are more commonly used for sex determination than any others (Fig. 1, Supplementary Table 3). Of the four species that were present in both our study and those from the literature, we were only able to identify the system of heterogamety in two species (R. japonica (East) and R. pipiens (West)) and the sex chromosome in one species (R. japonica (East)). In all cases they agreed. This literature search also revealed several cases of intraspecific polymorphism. R. temporaria mostly uses Chr01 throughout its distribution range, but Chr02 has also been shown to segregate with sex (along with Chr01) in one Swedish population[42]. In R. pipiens, both Chr02 and Chr05 have been described as sex-linked in different populations, representing an intraspecific turnover between lineages of Eastern and Western USA[43]. Other examples of intraspecific turnover have been described in Pelophylax porosus, which uses Chr05 in the Okayama race (subsp. porosus) and Chr03 in the Nagoya race (subsp. brevipodus), and R. japonica, which uses Chr01 in Western Japan but Chr03 in Eastern Japan. Interestingly, no linkage was found between sex and any of these two chromosomes in one R. japonica population from Akita (Northern Honshu), suggesting another, as yet undetected intraspecific turnover[20]. Finally, Chr08 is potentially involved in three turnover events in Glandirana rugosa, the first being a turnover to Chr08 from the ancestral state, and following that, two independent homologous turnovers from an XY to ZW sex-determination system[14].

**Rate and patterns of sex-chromosome turnovers in Ranidae**. We placed the sex-chromosome identities for each species onto a phylogenetic tree produced by combining data from Yuan et al.[44] and Pyron and Wiens[45]. The topology and node dates of our tree

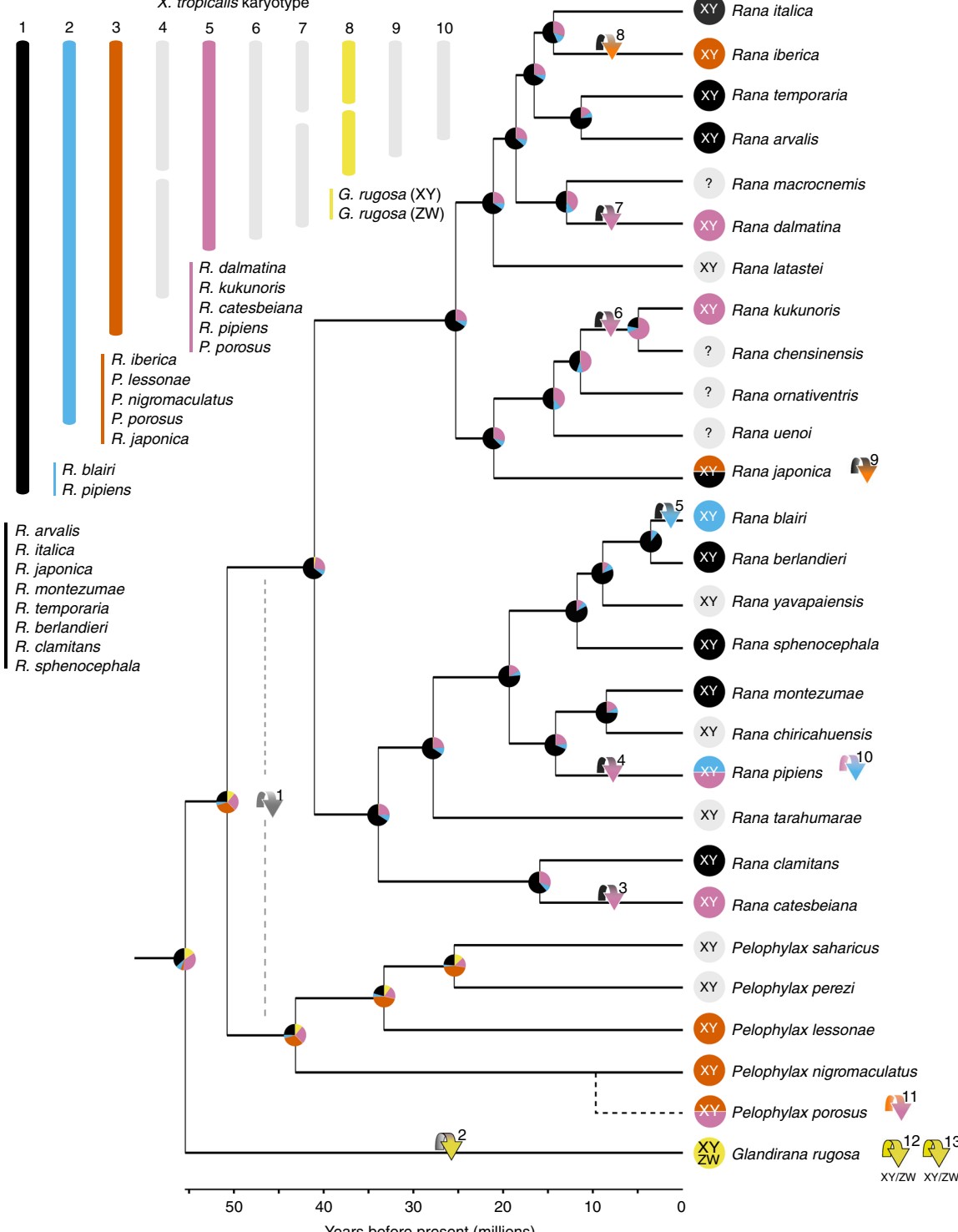

**Fig. 1** Sex chromosome turnovers across 28 true frog species. Sex-determination system and sex-chromosome identities come from both RADseq (Supplementary Table 1) and literature (Supplementary Table 2) data. Karyotype (top left, chromosomes not to scale) shows the number of species using each chromosome for sex determination and colours correspond to arrows, node pie charts and tips. Coloured arrows show the branch on which inferred turnovers occur based on the stochastic mapping analyses (Supplementary Fig. 8) and the pie charts at nodes represent the proportion of simulated trees in stochastic mapping with each of the states at that node. Tips with two colours represent intraspecific turnover events, with the transition described by the coloured arrow after the species name. Grey represents unknown sex chromosome identities in both tips and turnover arrows 1 and 2 and question marks at tips represent an unknown system of heterogamety. As there was not enough high-quality sequence information for *P. porosus*, it could not be included in the phylogenetic reconstruction or stochastic mapping. Its position here is inferred from that in ref. [45]

conformed very closely to those of both previous studies (Supplementary Fig. 7). We estimated ancestral states throughout the tree using stochastic mapping analyses, which allowed us to infer a minimum of eight sex-chromosome turnover events within the tree (Supplementary Fig. 8). When combined with the three intraspecific sex-chromosome turnover events in *R. japonica*, *R. pipiens*, and *P. porosus* and the two independent XY to ZW turnovers in *G. rugosa*, this equates to 13 sex-chromosome turnovers within approximately 55 M years (Fig. 1). In Supplementary Note 5, we outline an approach to calculate an approximate rate of transitions throughout our tree, which we envisage can be used to compare rates across different studies. Using this approach, we estimate a rate of approximately 0.02 turnovers per million years. This translates into 1 turnover for every 50 M years of independent evolutionary time (i.e. we would expect to see one turnover between two species which were 25 M years diverged from one another).

Although our ancestral states reconstruction was not able to confidently infer the state for the root of our phylogeny, it points strongly to Chr01 being the ancestral state for the *Rana* genus, which diversified approximately 40 M years ago. From this state, there have been two independent turnovers to Chr03 and four independent turnovers to Chr05. Ancestral state reconstruction in *Pelophylax* was based only on two species (i.e. too low to confidently infer ancestral states), but the sex-linkage of Chr03 in this clade indicates at least one more independent turnover event to Chr03 (plus one additional transition to Chr05 within *P. porosus*). Due to the ambiguous state at the root of the tree, the number and placement of these turnovers is unclear. Six hypotheses exist for the turnovers at this point in the tree (between nodes a, b and c, Supplementary Fig. 9); the most parsimonious (requiring the fewest turnover events) is represented on Fig. 1 (turnovers 1 and 2).

The number of times each chromosome was recruited for sex determination was highly variable. Chr01 was used most often (eight species), but in all cases was inferred to be the ancestral sex chromosome. Chr03 and Chr05 were both used by five species; however, Chr05 showed five independent recruitments, whereas only two or three (see turnover 1) were inferred for Chr03. Chr02 was used by two species, which represent two independent recruitments, and Chr08 was used by one species but, as mentioned above, is involved in three independent turnovers within the *G. rugosa* lineage.

As chromosome size varies greatly in Anuran genomes, we tested whether this pattern of sex-chromosome recruitment was consistent with expectations of a random model, taking into account the number of genes on each chromosome. Regression analysis showed no significant relationship between the number of times chromosomes had been recruited and the number of genes they contained ($R^2 = 0.028$, $p = 0.64$). By taking 1000 random samples of 13 genes (representing the 13 independent recruitment events) from the genome in silico, we show that the observed pattern of recruitment represents a significant departure from random expectation for Chr05, which has been independently recruited many more times than expected ($p < 0.001$;

Fig. 2). Furthermore, despite being the fourth largest chromosome in the genome, Chr04 was never found to be sex-linked in the species examined here. Taken together, these results clearly show a non-random pattern of recruitment of chromosomes for sex determination. Heterogamety shifts during transitions were not random either—eleven out of 13 transitions kept the ancestral system of male heterogamety. This differs significantly from binomial expectations when assuming an equal probability for both types of transitions ($p < 0.001$), but not when assuming a 3/4 probability of maintaining heterogamety, as expected under drift-induced transitions ($p \sim 0.1$).

**Patterns of sex-chromosome differentiation in *Ranidae*.** As our RADseq data provided multiple sets of sex-linked RADtags, we took the opportunity to examine their distribution along the sex chromosome, in an effort to elucidate the drivers of sex-chromosome turnovers among these species. As the previously anchored scaffolds were aligned to male linkage maps during the anchoring procedure, order within linkage groups was not reliable (because recombination is rare in male meiosis). We therefore created another linkage map combining data from female recombination patterns only across our six *R. temporaria* families (Supplementary Note 4, Supplementary Fig. 10). This linkage map contained 10,853 RADtag markers, confidently identified 13 linkage groups as expected (Supplementary Fig. 10) and showed the typical drastic difference between male and female recombination patterns previously observed for this species[41,46]. Expanding upon this previous work, we show explicitly that almost the entirety of the female linkage group falls into the non-recombining section of the homologous male linkage group (Supplementary Fig. 10), implying that almost the entire length of male chromosomes did not recombine in these six families (but recombined more than in females at chromosome tips). This map allowed us to order 328 Mb (7.31%) of the *R. temporaria* genome. We then aligned sex-linked RADtags from the seven species for which we could confidently identify a sex chromosome to these ordered scaffolds and plotted their relative positions on each linkage group (Fig. 3). In all cases, sex-linked RADtags mapped across almost the entirety of the linkage groups, with no clear islands of divergence that could be associated to a sex-determining region.

## Discussion

We have shown here that the rate of turnovers in true frogs is extremely fast, with at least 13 turnover events observed among 28 species within approximately 55 M years. In some cases, this has resulted in sex chromosomes that are likely less than 3.6 M years old (between *R. blairi* and *R. berlandieri*), or even younger in the case of intraspecific turnovers (e.g. *R. japonica*, *R. pipiens*, *G. rugosa* or *P. porosus*). This is, to our knowledge, the fastest rate of sex chromosome turnover observed to date. Several other systems have been identified as exhibiting frequent sex-chromosome turnovers, for example the genus *Oryzias* (seven different sex chromosomes identified among 14 species, which

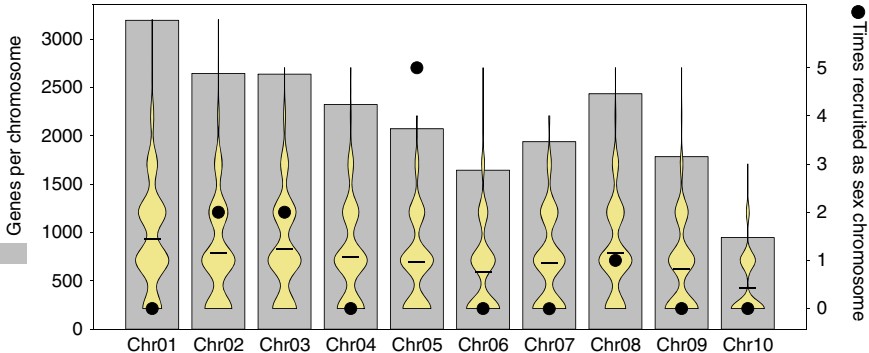

**Fig. 2** Non-random recruitment of chromosomes for sex determination. Grey bars represent the number of genes per *X. tropicalis* chromosome. Violin plots show the expected number of times each chromosome would be recruited if genes were recruited for sex determination at random (1000 replicates). Black dots represent the number of times each chromosome was observed as recruited from our data

diverged approximately 65 M years ago[12,25,47]) and Salmonids (six different sex chromosomes (often fused with autosomes) among eight species of salmon, trout, charr and whitefish[48,49]). However, these systems lack ancestral state reconstruction analyses and thus, the number and phylogenetic position of turnovers is not currently known. This unfortunately precludes a formal comparison between the rate observed in Ranidae to that of the other systems. However, we have devised an approach (Supplementary Note 5) that we envisage can be used to compare average rates of turnover among studies. Therefore, we hope that future characterization of transitions using ancestral state reconstruction in other systems will allow for this comparison, which would be highly informative for identifying the forces responsible for turnovers.

The rapid rate of turnover in Ranidae observed here would undoubtedly help to maintain sex chromosome homomorphy. This is, however, only a minimum estimate of turnover rate; it is likely that better knowledge of ancestral states would reveal even more turnovers than we are able to characterize here. Furthermore, our approach in principle cannot detect situations where the new sex determiner arises on the already sex-linked linkage group ('homologous' turnovers) unless it was accompanied by a change in heterogamety (XY to ZW), as was the case in *G. rugosa*. Also, we cannot rule out that several turnovers have occurred on the same branch in our phylogeny. Both phenomena would act to increase the observed rate of turnovers.

In order to infer the major forces driving the high rate of turnovers in this system, we asked whether patterns of heterogamety were conserved during turnovers. Of the 13 turnover events observed here, 11 of them were XY to XY, with the two XY to ZW turnovers both being in *G. rugosa*. Even if one or more of the four species for which no sex-determination system could be found with either RADseq or in existing literature (*R. macrocnemis*, *R. chensinensis*, *R. ornativentris* and *R. uenoi*) were in fact ZW, this would still leave a clear bias towards XY heterogamety in this family. This bias runs against expectations from models where either sex-ratio selection or sex-antagonistic selection drive turnovers[31–33,50]. Two counterarguments could be made here. First, mechanisms are not mutually incompatible; if sexual antagonism was driving turnovers but mutationally loaded Y chromosomes cannot be fixed, then heterogamety would be preserved as well, but the causal force would not be the deleterious load. Second, sex-antagonistic selection could also bias turnovers towards XY systems in Ranidae: given the much reduced recombination in males, a male-determining mutation occurring on an autosome is more likely to benefit from linkage with a sex-antagonistic gene than is a female-determining

mutation[50,51]. Although both arguments are correct in principle, the point must also be made that sex-antagonistic selection is not expected to generate the continuous cycles of turnovers as documented here. An autosomal male-beneficial mutation might indeed trigger an initial XY-to-XY transition but should strongly oppose any further transition once sex-linked[34,35]. Furthermore, genomic investigations do no support a role for sex-antagonistic genes in sex-chromosome evolution in *R. temporaria*[52]. Overall, data in hand seem more consistent with a role for mutation-load selection or possibly drift (or a combination of the two). It is worth noting that the only known exceptions to the general pattern (both in *G. rugosa*) have explicitly been assigned to sex-ratio selection: the ZW races likely result from two hybridization events between highly diverged XY lineages of *G. rugosa*, and lab crosses between these parental lineages indeed show a male bias in the progeny, which should suffice to select for a dominant feminizing mutation[14]. If this interpretation is correct, then transitions left to be explained all keep the original pattern of male heterogamety, which further increases the likelihood of the mutation-load model over the genetic-drift model.

If mutation load is indeed the driving force for the preservation of XY heterogamety in Ranidae, then the sex-specific patterns of recombination might help explain why the rate of sex-chromosome turnover is so high in frogs relative to many other taxa. Frogs exhibit extreme patterns of heterochiasmy (different recombination rates between males and females): the ratio of female-to-male map length has been estimated at ~70 in *R. temporaria* based on microsatellite markers[53], a remarkable result given that values normally range 0.5–2[54–56], with a few outliers in fishes (up to 8.26 in the Atlantic salmon *Salmo salar*[57] and down to 0.135 in the Japanese flounder *Paralichthys olivaceus*[58]). From our present RADseq data, this strong sex difference stems from the fact that all chiasmata in male meiosis form at chromosome extremities (Supplementary Fig. 10). A similar pattern was recently described in stickleback, although the clustering of chiasmata towards the telomeres was not as extreme[51]. As meiotic recombination depends on phenotypic sex (not on genotypic sex), X and Y chromosomes may still recombine occasionally in the rare sex-reversed XY females that occur when alleles at the sex locus show incomplete penetrance[59], preventing X–Y differentiation. This possibly accounts for our inability to identify sex-linked markers in some of our samples. However, recombination will immediately cease along almost the entire length of the X and Y chromosomes as soon as full-penetrance alleles are fixed, which is clearly the case in some *R. temporaria* populations[42,60,61] as well as in populations of other Ranid species that display extensive X–Y differentiation (Fig. 3). The sudden linkage of a very

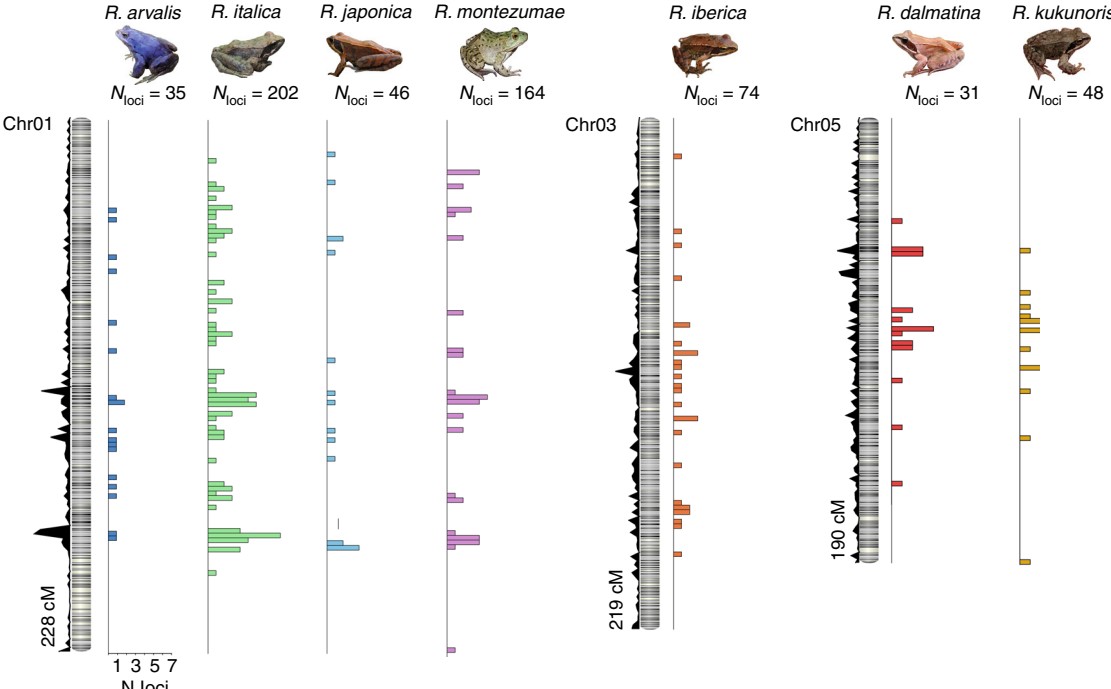

**Fig. 3** Chromosome-wide distribution of sex-linked RAD markers in seven true frog species. Density plots to the left of linkage maps show the marker density along the linkage group. Coloured bars are histograms of the number of sex-linked markers along the linkage group, with bins of 2 cM. Picture credits: Jan Jezek (*R. arvalis*), Astolinto (*R. italica*), Yasunori Koide (*R. japonica*), Jim Rorabaugh (*R. montezumae*), DLJ (*R. iberica*), NR (*R. dalmatina*), AB (*R. kukunoris*)

high number of genes is expected to generate particularly strong Hill–Robertson interferences. As a result, deleterious mutations will immediately begin to accumulate, and will do so along the length of the entire chromosome (except at the tips). This process is in stark contrast with the progressive expansion of the non-recombining region outwards from the sex determiner posited by the canonical model of sex-chromosome evolution, and does not require the appearance of sexually antagonistic genes to favour recombination arrest along the sex chromosome. In the absence of sexually antagonistic alleles, the sex-limited chromosome is expected to very quickly become less fit than any potential proto-sex-chromosome that would arise. It then follows that mutation-load selection would quickly favour a turnover as soon as a new option for sex determination is available (a particularly efficient version of the 'hot-potato' model of sex-chromosome turnover[35]). It would be interesting to quantify deleterious mutations on differentiated Y chromosomes and examine sex-chromosome differentiation and turnover in other systems with similar patterns of heterochiasmy, to see if this hypothesis holds.

A central question of this study was: are some chromosomes more likely to be recruited for sex determination than others? We show here that recruitment departs significantly from random expectations in true frogs. Of the 13 chromosome pairs, only five are used for sex determination, with three of them being recruited multiple times independently—most notably Chr05 which has been recruited an astonishing five times. Chr01, used as sex chromosome in the majority of species investigated here, has also been co-opted independently in species from two highly diverged clades of frogs (Hylidae and Bufonidae)[62]. This raises the question: why are some chromosomes, for instance Chr01, Chr03 and Chr05, more likely to be recruited for sex determination than others? Is it that some chromosomes are better at sex determination than others[63]? One explanation for this could be the presence of particular genes on these chromosomes that play

important roles in the sex-determination cascade. Indeed, several genes (or their paralogs) have been recruited independently more than once as master sex-determiners, suggesting that they are particularly good at fulfilling this role. These so called 'usual suspects'[64] include *Sox3*, which maps to Chr08 (involved in three transitions within *G. rugosa*). *Sox3* determines sex in several medaka (*Oryzias*) species[38] and is a strong candidate in *G. rugosa*[65], while its paralog *Sry* is the master sex-determining gene in therians[66]. The transcription factor *Dmrt1*, which plays a crucial role in the sex-determination cascade throughout all animals (and maps to Chr01), has been independently co-opted for sex-determination in birds[67] and some fishes[68] and it is a strong candidate in *R. temporaria* and in some tree frogs (*Hyla*)[69,70]. Furthermore, its paralogs *DmY* and *DmW* determine sex in some medaka fishes (*Oryzias*) and clawed frogs (*Xenopus*), respectively[38,71]. These paralogs have functions related to those of their parent genes (e.g. the female determiner *DmW* is thought to block the binding of *Dmrt1* to its target sequence, thus suppressing the male determining cascade[72]) lending even more credence to the idea that the specific functions of these genes make them particularly proficient at the role of master sex-determiner. Interestingly, Chr05 in frogs harbours *Foxl2* (forkhead box L2), also one of the most important genes in the vertebrate sex-determination cascade[73]. It codes for a transcription factor essential for ovarian development and has also been implicated in the suppression of testis formation. *Foxl2* interacts directly with *Dmrt1*; the male-determining *Dmrt1* allele blocks the expression of *Foxl2*, and in turn the development of ovaries, thus producing males. Indeed, studies in both tilapia[74] and zebrafish[75] have shown that the knockdown of *Foxl2* results in female to male sex reversals. Thus, a plausible hypothesis for the recurrent recruitment of Chr05 for sex-determination is that a loss-of-function mutation in *Foxl2* uncouples its activity from that of *Dmrt1*, allowing it to take over as the primary male determining gene.

Accordingly, Chr05 might not just be good at the role of sex chromosome, but more specifically it might be good at taking that role from Chr01.

Still, we cannot rule out a translocation of the same sex determiner to new chromosomes. In the case of the jumping sex-determining gene in salmonids, Lubieniecki et al.[48] found that three chromosomes, all of which are sex-linked in some populations, harboured sequences that were homologous to the regions flanking the sex determiner. These regions facilitated the translocation of the sex determiner to these locations, producing a recruitment bias towards some chromosomes (as well as a pattern of strongly preserved male heterogamety) similar to the result seen here. We note, however, that the two transcription factors identified as serious candidates in our dataset (respectively Sox3 in G. rugosa and Dmrt1 in R. temporaria) did not jump in this specific instance (Dmrt1 is autosomal in G. rugosa, and Sox3 is autosomal in R. temporaria). We also note that convergent recruitment of the same genes should not be considered as an alternative to the hot-potato model to account for the maintenance of heterogamety. We rather see the accumulation of deleterious mutations as the ultimate cause (selective force), and the convergent recruitment of genes from the sex-determination cascade as the proximate cause. Convergence per se does not impose maintenance of heterogamety, as mutation of any gene in the male cascade can be either masculinizing or feminizing (a mutation upregulating Dmrt1 expression will be masculinizing, while a mutation downregulating its expression will be feminizing). Future work should concentrate on validating candidate sex-determining genes in this family.

In conclusion, we show here through the combination of novel and existing data that sex-chromosome turnover among frogs occurs at a very high rate. This may be a result of the extreme heterochiasmy that characterizes frogs, which should induce a rapid accumulation of deleterious-mutation load, and thereby quickly select for a turnover (a frog version of the 'hot-potato' model[35]). This would account for the consistent maintenance of male heterogamety despite frequent transitions. Intriguingly, these turnovers converge on only a few chromosomes, implying that they are particularly adept at the role of sex determination, perhaps due to specific genes that they harbour.

## Methods

**Sample collection and RADseq library preparation.** In total, we generated RADseq data for 19 species of true frogs (Family: Ranidae). The taxonomy is still debated within this family[76], so here we opt to use the simpler early nomenclature also recently used by Yuan et al.[44]. Under this nomenclature, these species fall into two genera, Rana s.l. (sometimes split into Rana and Lithobates) and Pelophylax. We obtained male and female samples of each species (mean $N_{FEMALES} = 22.2$, $N_{MALES} = 21.7$), totalling 736 individuals (Supplementary Table 1).

Samples collected for this study were done so under the following permits: R. berlandieri, R. chiricahuensis, R. tarahumarae and R. yavapaiensis samples were collected by M.J.S. under the authorities in Arizona Revised Statutes, Title 17, and under Section 6 authorities granted by the Endangered Species Act. R. sphenocephala samples were collected by L.N.B. under permits from the FL Fish and Wildlife Conservation Commission (LSSC-09-322). R. montezumae samples were collected by C.M.G. under permits from Secretaría del Medio Ambiente y Recursos Naturales (SEMARNAT) (SGPA/DGVS/02919/15). R. latastei samples were collected by G.L., W.J.M. and K.G. under permits from Ufficio della natura e del paesagio, Ticino. R. italica was collected by A.R. under permits issued by the Italian Ministry of Environment (0003951/PNM of 03/03/2015). R. dalmatina was sampled by D.C. with permits from the Ministry of Environment, Italy (Prot.0007727). R. iberica was sampled by A.G.N. under permits from the Picos de Europa National Park and the Principality of Asturias (Spain). P. perezi was sampled by I.M.S. under permit 10/016997.9/13 from Consejería de Medio Ambiente, Comunidad de Madrid (Spain). P. saharicus samples were collected by PAC under permits from Haut Commissariat aux Eaux et Forêts et à la Lutte Contre la Désertification. No permits were needed for sampling of R. uenoi (according to Korean law) or for R. japonica, R. ornativentris, R chensinensis and R. kukunoris (according to Chinese law).

On collection, individuals were also phenotypically sexed, either morphologically, in many cases using the presence or absence of nuptial pads or vocal sacs, or by dissection and examination of gonads[77].

DNA was sampled from collected frogs by fixing tissue (tail clips, toe clips, leg muscle, whole frogs) in 70–90% ethanol or tissue buffer, or by taking buccal cell swabs. We then extracted DNA using the Qiagen® DNeasy® Blood & Tissue Kit and produced double-digest RADseq libraries for each species according to the protocol described in Brelsford et al.[78]. In brief, we digested genomic DNA using restriction enzymes SbfI-HF and MseI and then ligated adapters containing unique barcodes of 4–8 bases separately to each sample before amplifying the libraries using PCRs of 20 cycles. We then pooled PCR products and performed a gel-based size selection, isolating fragments of approximately 400–500 bp. The finished RADseq libraries were single-end sequenced on an Illumina HiSeq 2500.

**Raw data processing and SNP calling.** Illumina raw reads were quality checked using FASTQC v0.10.1[79] and were demultiplexed by individual barcode using the process_radtags module of STACKS v1.48[80]. STACKS modules Ustacks, Cstacks, Sstacks and Populations were then run separately. For all datasets, we ran initial tests using multiple values for the core STACKS parameters (Ustacks: -M, -m and Cstacks: -n), but in all cases, the default values provided a good balance between data quantity and quality, thus these defaults were used for all species. In the final analyses, SNP markers were retained if they were present in at least 75% of both males and females. Additionally, we specified a minimum minor allele frequency of 0.05 and a maximum observed heterozygosity threshold of 0.75. The latter acts to remove over-merged stacks resulting from repetitive elements, as each paralogous copy of the repeat will look like a separate allele in such stacks, these false-loci are likely to have a much higher number heterozygous calls across individuals compared to either Hardy–Weinberg expectations (Population data) or Mendelian segregation expectations (Family data, except for loci homozygous for different alleles between the two parents, which are thus not sex-linked). Note that there is the possibility of removing sex-linked markers using such a filter if the number of the heterogametic sex makes up >75% of the samples. However, this was not the case for any of our sample sets. The number of high-quality SNPs retained per dataset ranged from 2351 to 71,462 (Mean = 31,515, SD = 22,021).

**Identifying sex-linked markers.** Final SNP datasets for each species were bioinformatically screened for signatures of sex-linkage, for both male (XY) and female (ZW) heterogametic systems, using the three approaches previously described in Brelsford et al.[81]. These approaches are described briefly below. For the sake of simplicity, we refer to only XY systems, but in all cases the reverse is true for ZW systems.

Approach one screens for loci with a Y-specific SNP. In an XY system, these SNPs will be present at a frequency of 0.5 in males and 0 in females. We therefore consider a locus to be sex-linked if the X-allele frequency is ≥0.95 in females and ≥0.4 and ≤0.6 in males, allowing a margin of error for a few incorrect genotype calls or phenotypic sex assignments.

Approach two also screens for Y-specific alleles, but instead uses the differences in patterns of heterozygosity between males and females. Under this approach, a locus is considered sex-linked if it is homozygous in all females and heterozygous in at least half of the male samples. This method is likely to return many of the same sex-linked markers as the first approach but is more stringent in its conditions. For example, an allele frequency of 0.5 in males may be possible by chance if the sample set is small, or if there is population structure within males leading to consistently fixed loci among them. Thus, it is expected that approach two will be less sensitive to false positives in such situations.

Approach three screens for sex-limited RADtags that are specific to the Y chromosome. In this approach, RADtags are assessed for their presence or absence in one sex and are considered sex-linked if they are completely absent in the homogametic sex and present in at least half of the heterogametic sex. Several mechanisms can result in such sex-specific loci: a deletion on the X chromosome, a mutation in the restriction site on the X chromosome (leading to a sex-specific null allele), a mutation on the Y creating a novel Y-specific restriction site, or under-merging of a RADtag in Stacks due to high divergence between X and Y alleles. In the latter case, only the Y allele will be sex specific and thus, can still be used as a sex-linked marker.

In several species, samples were collected from multiple populations or lineages. It was therefore necessary to split these data into subsets and screen each one for sex-linked markers separately (see Supplementary Table 1), in order to control for the potential noise produced by population structure.

**Identifying the sex chromosome.** In order to identify the sex chromosome in each species, it was necessary to locate the RADtags identified as sex-linked on a reference genome. The most reliable chromosome-level assembly in frogs is that of X. tropicalis[82] and, being an outgroup, it provides a good reference with which to unambiguously name chromosomes. However, due to the amount of divergence between Ranidae and X. tropicalis (210 M years[83]), mapping rates of 92 bp RAD-tags to this reference genome are almost zero. We therefore anchored as many scaffolds as possible from an existing fragmented R. temporaria genome assembly[41] to the X. tropicalis assembly. We created linkage maps using RADseq data from six

*R. temporaria* families from Rodrigues et al.[61] (60–90 offspring per family) and used these, alongside *Nanorana parkeri* and *X. tropicalis* genomes, to assign scaffolds to their homologous *X. tropicalis* chromosome (henceforth referred to as chromosome-assigned scaffolds). Note that synteny is known to be highly conserved between *N. parkeri*, *X. tropicalis* and Ranidae[46,62,84,85]. Linkage mapping and assignment of scaffolds to chromosomes is detailed in the Supplementary Note 4 and Supplementary Fig. 3.

RADtags identified as sex-linked were aligned to the chromosome-assigned scaffolds in order to identify the chromosome to which they belonged. All alignments mentioned above were performed using blastn v2.3.0[86], with hits only retained if their e-value was below $1 \times 10^{-20}$ and, in the case of multiple matches, at least five orders of magnitude lower than that of the next best hit (to account for the repetitive nature of Anuran genomes[84]).

We combined our new data with already-published sex-chromosome identities for other species from the literature (Supplementary Table 3). However, as many studies use cytogenic methods, which are not completely reliable in the case of such homomorphic sex chromosomes, and to avoid confusion from ambiguous chromosome naming, we included only studies that used sex-linked allozymes to identify the sex chromosome. The location of the enzymes used could then be reliably confirmed by searching them against the well annotated *X. tropicalis* genome (Xenbase.org[82]). Of the 19 species for which new data were generated in this study, five were present in the retained studies from our literature search (*R. arvalis*, *R. japonica*, *R. berlandieri*, *R. sphenocephala*, *R. pipiens*).

**Validating sex-linked marker and sex-chromosome identification.** When screening large SNP datasets for sex-linked loci, it is imperative to account for the possibility of false positives, i.e. loci which fit the expected pattern of sex-linkage due to artifacts or simply by chance, without being truly sex-linked. Here, we used a permutation approach to increase the confidence around sex-linked marker and sex-chromosome identification, while accounting for the sample size, sex-skew and biological characteristics of each dataset. For a given dataset, we first used true phenotypic sex assignments, and the three approaches detailed above, to find putative sex-linked markers. We then calculated the empirical null distribution for the rate of false positive discovery by permuting male and female assignments randomly across the sample set 1000 times, each time screening for sex-linked markers with all three approaches. We retained only sex-linked marker sets where the number found using the real sex assignments fell outside of the 99th percentile of the null distribution, i.e. had a probability of <0.01 of occurring by chance.

We also used this permutation approach to identify which dataset properties were the most important predictors of the number of sex-linked markers or false positives found. To do this, we extracted various subsamples from the already published RADseq dataset from *R. arvalis*[81]. We chose this dataset for several reasons: first, the sex-linked markers in this dataset (identified using the same three approaches as used here) have been validated via the use of PCR. Second, the original sample size (29 males and 19 females) was large enough to allow for several rounds of downsampling. And third, exploratory PCA clustering of individuals using only the sex-linked markers shows evidence for two major Y haplotypes present among males (referred to here as Hap 1 and Hap 2). This dataset therefore allowed us to test four potential predictors for the success of sex-linkage screens: (1) the effect of overall sample size (via a series of subsamples from 19 males (M), 19 females (F) down to 5 M, 5 F), (2) a skew in sample size towards males (18 M, 9 F), (3) a skew towards females (9 M, 18 F), and (4) the effect of having multiple sex-chromosome haplotypes in the same dataset (6M$^{Hap1}$, 6M$^{Hap2}$, 12 F). The latter factor has not previously been considered when assessing the stringency of screens for sex-linked markers.

Putative sex-linked marker sets that fell outside of the 99th percentile of the empirical null distribution were then used in the next step of the analyses, finding the location on the reference genome using the chromosome-assigned scaffolds. However, as the presence of false positives in sex-linked marker sets could not be ruled out, and because the amount of divergence between the target species and *R. temporaria* chromosome-assigned scaffolds was often high (5–41 M years), we expected some noise when aligning markers stemming from either false mappings, mapping of false positive sex-linked loci which are actually autosomal, or true translocations that have occurred since the divergence of the target species and the *R. temporaria* reference genome. Such errors could lead to false signals of sex linkage for chromosomes other than the true sex chromosome. Thus, it was also necessary to calculate the expected false-positive mapping rate, per dataset and per *R. temporaria* chromosome. To do this, we first mapped the putative sex-linked markers for a given dataset to the ordered *R. temporaria* scaffolds. We then took 1000 random subsamples of RADtags from the Stacks catalogue of loci for that dataset, with each subsample being of the same size as the set of sex-linked markers. Each subsample was mapped to the chromosome-assigned *R. temporaria* scaffolds to give a distribution of the expected mapping rate for each chromosome.

For final sex-chromosome identification, we required at least ten sex-linked markers to align to a single chromosome and the number of sex-linked marker alignments to fall outside of the 99th percentile of the random marker alignment distribution for that chromosome.

**Assessing patterns of sex-chromosome turnover.** In order to test whether sex-chromosome recruitment was random, or if some chromosomes are more likely to

be recruited, it was necessary to place all sex-chromosome identities onto a phylogenetic tree. A phylogeny was therefore produced using a combination of data from two previous phylogenetic studies[44,45]. We used all ten loci in ref.[44], which included four mitochondrial genes (*12S*, *16S*, *cytb* and *nd2*) and six nuclear genes (*rag1*, *rag2*, *bdnf*, *slc83a3*, *tyr* and *pomc*). Of the 28 species in our study, 22 were included in ref.[44]. For the remaining six (*R. italica*, *P. lessonae*, *P. perezi*, *P. porosus*, *G. rugosa* and *P. saharicus*), we used data for *12S*, *16S* and *cytb* from ref.[45]. Genbank accession numbers for all sequences used can be found in Supplementary Table 4.

Sequences were first aligned separately for each gene, using MUSCLE[87] as implemented in Geneious v11.1.5[88]. Alignments were then manually trimmed, excluding regions with high amounts of missing data or ambiguous alignment. The ten trimmed gene alignments were used to make a supermatrix, which was partitioned according to gene and codon position. Best-fit substitution models were then found for each partition using PartitionFinder2 (Supplementary Table 5)[89–91]. Yuan et al.[44] showed that maximum parsimony, maximum likelihood and Bayesian inference all produced very similar trees, thus here we used only a Bayesian approach as implemented in BEAST 2[92]. We also incorporated the four fossil-based node calibrations used in Yuan et al.[44] in order to date the tree, using a relaxed uncorrelated clock[93] and a Yule birth–death model. Trees were calculated using MCMC chain lengths of 100 M, with sampling every 10,000 iterations, ensuring that the effective sample size (ESS) of each parameter in the model was above 200.

Using this phylogenetic framework, we inferred ancestral states using a stochastic mapping approach, implemented in the R package Phytools v0.6-44[94]. The best model for transition rates between these states was identified by comparing the likelihood scores (using Akaike Information Criterion (AIC)) for three different transition rate models, equal rates (ER), symmetrical (SYM), all rates different (ARD). To reduce noise and increase the power of this analysis, we assumed that only the five chromosomes that had been confidently identified as a sex chromosome in at least one of our species could determine sex; thus we considered five possible states in the reconstruction. Because of sex-chromosome polymorphism within some species, we coded the states at the tips as a probability matrix, giving the different states equal probabilities in that species. For species where we could not detect the sex-determination system, all five possible states (sex-chromosomes) were given equal probabilities. Using these inputs, we reconstructed ancestral states along each branch for 1000 simulated trees. At any given point in the tree, the most likely state was the one with the highest number of simulated trees supporting it at that position, thus, turnovers could be placed on branches where there was a switch in the state with the most support. We also repeated this analysis with all ten possible chromosome states included, which confirmed that the use of only five states did not bias ancestral states towards any particular chromosomes.

Finally, using the number of turnovers and the identities of the ancestral/derived chromosomes for each one, we tested whether or not the observed pattern might be expected under a random recruitment model. We first used an ordinary least-squares regression to test for a correlation between the number of times a chromosome had been recruited and the number of genes on that chromosome (taken from the homologous chromosome of *X. tropicalis*). We then simulated the random expectation for the number of recruitments per chromosome by repeatedly (100 replicates) sampling a subset of *N* genes ($N$ = number of sex-chromosome recruitments observed) from the genome and comparing these distributions to the observed number of recruitments for each chromosome.

**Assessing patterns of sex-chromosome divergence.** As our RADseq data provided multiple sets of sex-linked RADtags, we took the opportunity to examine their distribution along the sex chromosome, in the hope of elucidating the drivers of sex-chromosome turnovers among these species. Recombination occurs much less frequently in male frog meiosis, and because Y chromosomes are almost always found only in males, we hypothesized that, instead of the Y chromosome diverging from the X only at a small non-recombining region around a sex-determining, or sex-antagonistic genes, the frog Y chromosomes would diverge from the X across almost their entire length, except for the tips of chromosomes where recombination is known to occur[41]. To test for this expected pattern of divergence between gametologs, we first produced an additional linkage map in which we combined female-informative markers from all six *Rana temporaria* families into a single map using Lep-Map3[95] (see 'Linkage mapping' in Supplementary Note 4). While this approach resulted in a map with fewer markers than the six separate male maps produced with MSTmap, using only female linkage information provided more accurate relative marker positions within linkage groups. We used this map to order *R. temporaria* scaffolds along each *X. tropicalis* chromosome, and hereafter refer to these scaffolds as 'ordered scaffolds'. These could then be used to find the relative location of sex-linked markers on the sex chromosomes identified across different species by simply aligning sex-linked markers to the ordered *R. temporaria* scaffolds using blastn.

**Code availability.** All custom scripts and functions used can be found in the github repository https://github.com/DanJeffries/My_misc_scripts. Ipython notebooks detailing the use of these scripts in data exploration and sex-linked marker screens

for all species can also be found on Figshare (https://doi.org/10.6084/m9.figshare.6949331), please refer first to the README.txt distributed with these notebooks.

## Data availability

All sequencing data used in this study can be found (demultiplexed by sample) on the NCBI Sequence Read Archive under the Bioproject accession PRJNA478189. Stacks intermediate files for each species, linkage maps and alignment and tree files for the phylogeny can be found on Figshare (http://dx.doi.org/10.6084/m9.figshare.6949331) along with fasta files containing putatively sex-linked markers where identified.

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

## Acknowledgements
We thank E.A. Luna, J. Leuenberger, A. Athanasiades, J. Buser, A.D. King, A.H. McCall, S.J. Sferra, M. Biaggini, G. Sánchez-Montes and Dario Dominighetti for their help with collection and processing of samples. We also thank Philippe Geniez and the collection B. E.V. of the C.N.R.S. for the *P. saharicus* samples, Nicolas Salamin for advice on phylogenetic analyses and Barret Phillips for comments on the manuscript. This work was primarily funded by two grants (31003A_166323 & CRSII3_147625) awarded by the Swiss National Science Foundation to N.P. All computations were performed at the Vital-IT (http:// www.vital-it.ch) Center for high-performance computing of the SIB Swiss Institute of Bioinformatics

## Author contributions
N.P. and A.B. conceived the project and were involved with study design alongside D.L.J. and G.L.. D.L.J., G.L., W.J.M. and R.S. carried out the DNA extractions and RADseq library preparations. D.L.J. and G.L. performed the analyses. D.L.J., G.L. and N.P. wrote the manuscript. I.M., I.S., W.J.M., M.J.S., J.F., N.R., C.D., K.G., C.M.G, I.M.S., A.B., D.C., A.R., R.P.O., L.N.B., A.G.N., P.A.C. and S.Z. provided the samples and commented on the manuscript.

## Additional information

**Competing interests:** The authors declare no competing interests.

