## [Peer Review File · Nature Communications]

Reviewers' comments:

Reviewer #1 (Remarks to the Author):

Jeffries et al identify male and female heterogametic sex chromosomes across multiple frog species, with the aim to characterize the rate of turnover between XY and ZW chromosomes and establish whether some autosomes are predisposed to be sex-linked. Identifying the factors driving sex chromosome turnover is a hot topic in evolutionary biology yet identifying sex chromosomes in groups with high rates of turnover is not a trivial task. The manuscript is extremely well written and I applaud the authors for their clear and transparent methods. This is the first paper to my knowledge to attempt to rigorously distinguish between difference scenarios promoting sex chromosome turnover and will likely be of broad interest to many people. Please see my detailed comments below.

For the ancestral state reconstruction, the authors only consider 5 possible states. Therefore, for species where sex-determination is unknown, only 5 possible chromosomes were permitted to be considered for sex-linkage. This seems rather strange to me and might instead bias the analysis to find that the same chromosomes are more often recruited for sex-linkage than in reality. It would be more appropriate not to impose any assumptions on the data at this point. Indeed, from Fig S6, although it is not significant, it seems there might be evidence that Chr 7 is sex-linked in *R. pipiens*. This would then be 6 possible states. The authors should redo this allowing all chromosomes to be potentially sex-linked. Given this analysis forms the basis for the central question of the study, rerunning it without a priori assumptions is extremely important. Are the results comparable?

In the discussion, the authors mention that the pattern of sex chromosome turnover is consistent with a mutation load or drift hypothesis. The majority of the discussion (and abstract) is then focused on the mutation load hypothesis. The justification seems to be that male heterochiasmy would promote rapid accumulation of deleterious mutations. However, occasional recombination has been shown in this group and surely this would purge the deleterious mutations? There needs to be more focus on the impact of this in the discussion. To me, it makes the mutation load hypothesis rather unlikely. What would be mutation and recombination rates need to be in order to for this scenario to be plausible?

Given the importance of correctly identifying sex chromosomes for the conclusions of this paper, the authors do a fantastic job in investigating inconclusive patterns of sex linkage. In Fig S2 they show that the signal of XY and ZW sex-linked markers on the same chromosome is due to paternal X-specific SNPs. However, they show this only for *R. montezumae* (panel e). Can they also show the same is true for *R. italica*, *R. chrichahuensis* & *R. kukunoris*? Or are there any independent lines of evidence supporting an XY system in *R. italica*, *R. chrichahuensis* & *R. kukunoris* as in *R. arvalis* where there is PCR validation? Given the uncertainty over these species, it would be reasonable to conduct additional PCR validation tests.

The evidence in support of the expected patterns of heterogamety under drift are detailed in a paper that is not publicly available. The authors should give further detail here to explain

this hypothesis.

Fig S3 panel c). Checking the effect of sample size on false positive rate is great. However, this would be more appropriate if it was conducted on the Y1 subsample, so the starting point is significant evidence of XY. I am not sure in its current form that the figure supports the author's conclusion that the tests can confidently identify sex-linked markers down to sample size of 7M and 7F because 1) the ZW markers are false positive, 2) everything is significant at all sample sizes. A similar figure testing the effect of unbalanced male and female sampling for the identification of ZW and XY markers would also be beneficial here.

L136 Missing 'more' likely

L493 Can the authors provide extra explanation for why these parameters will ensure over-merged loci resulting from repetitive elements are removed. They seem rather arbitrary at the moment.

L504 Font changes in this paragraph

Fig S6 What is the significance of the dotted box around Chr 2 in *R. latastei*? Why does *P. perezi* not have one?

Fig S8 What is the significance of the * next to the species name? This is not explained in the figure legend.

Reviewer #2 (Remarks to the Author):

The authors characterize the sex chromosomes of many ranidae species, and find a very high rate of turnover, potentially explaining their homomorphy despite the lack of recombination in males. I think that these are very interesting and well written results based on a sound analysis, but have comments about the interpretation of some of the patterns. I also feel that previously reported findings could be made clearer throughout the text.

1. Maintenance of male-heterogamety and the hot potato model:

The authors view the maintenance of male heterogamety as evidence for the hot potato model. However, this is also expected if the same gene is determining sex at different locations, and if the same genes are recruited convergently for sex determination. Although I agree that the lack of recombination in males makes ranidae likely to suffer from a quick accumulation of deleterious mutations on the Y, the authors don't have any evidence of it happening in the species under study. I think that the discussion of these patterns needs to be a bit more balanced, and other models that predict the maintenance of male-heterogamety mentioned.

2. Novelty of the patterns described:

The same group had previously characterized the sex chromosomes of *R. arvalis* (XY, chr1),

yet this species seems to count towards the 20 frog species for which "the sex determination system and identity of the sex chromosome were previously unknown". Similarly, *R. berlanderi*, *R. pippiens*, *R. sphenoccephala*, *R. japonica* were at least partly characterized in the literature (and in several cases the sex chromosome assignment shown in figure 1 came from the literature and not from the new data, which did not have enough power to detect the SD chromosome).

Of the 7/20 species for which the sex chromosome could be determined, 2 were confirmations.

This means that at least 7 of the transitions were reported from the literature.

I think it should be clearer in the main article text and figures what is new data and what is previously reported.

3. *Pelophylax* and *Glandirana* have different names in figure 1 and table s3.

Reviewer #3 (Remarks to the Author):

The manuscript reports on the rate of sex chromosome turnover among frogs. For this study RADtags were produced from 20 species of the frog family Ranidae to add additional data for species where the sex chromosome system in this family was not already known. In total, except for four species they obtain information for the 28 species that they selected. Plotting the sex chromosome data on the phylogenetic tree revealed the highest rate of sex chromosome turnover in vertebrates known so far. Their data are discussed in the framework of major hypotheses of sex chromosome evolution and give important insights into the evolutionary forces that shape this process. The paper is well written and is of outstanding importance for the research field. It should be of interest also for scientists that work in related areas. In general, it is very well suited for publication in *Nature Communications*. The experimental and bioinformatics analysis were all done with state of the art technology and are described in sufficient detail. The important drawback of the unavailability of a chromosome-anchored high quality reference genome for this family of frog species is elegantly circumnavigated with clever usage of trans-mapping approaches.

There are, however, several points that need to be considered, before this paper can be published:

- Lines 198 ff: The lack of information from the RAD-tags in some species raises concern. It is pretty likely that for *R. macrocnemis*, *chinensis*, *ornaventris* and *uenoi* not enough individuals were analysed, probably due to unavailability. Gamble et al. may just have been lucky with low numbers of individuals or the enzyme used. In other studies the experience is that increasing the sample size improves the outcome of RAD-tag analyses. Another possibility is that these 4 species have either molecularly undifferentiated sex chromosome (remember *Fugu*!) or have polyfactorial genetic or environmental sex determination. Anyway, it is difficult to understand why these species were then included in the overall

analysis, in particular in Figure 1, which gives the main message. There is no mention of this issue in the whole paper and I would see it as improvement to delete them.

- Line 293: *G. rugosa*, not *R. rugosa*
- Lines 335 – 362: In this context it is of interest to compare the situation in the Ranidae not only to the genus *Oryzias*, but also to Salmonids and the Geckos. Both groups are also well-known for high sex chromosome turnover and the reader can expect to be told how these compare in terms of frequency and time with the frogs. In particular, the finding of 19 transitions in the geckos compared to the 13 in Ranidae needs discussion.
- Lines 386 – 403: The authors use what they call “extreme patterns” of heterochiasmy as argument against the importance of sexually antagonistic genes in the process of sex chromosome evolution and as support for their “hot potato” model. Here the reader will expect a comparison of the extent of heterochiasmy in the Ranidae with the rates known for most mammals and fish, e.g. zebrafish. According to the Haldane-Huxley rule achiasmy occurs in the heterogametic sex. Is there an inversion of the pattern in the W/Z *G. rugosa*?
- Lines 417 – 439: This part should be shortened considerably, because it is highly speculative. The given examples of candidate genes on the most frequently used chromosomes are clearly very suggestive, but considering that hundreds of genes will be located in the regions identified by the RAD-tags, many other candidate genes will exist. First, not always the usual suspects can be made responsible (see IGF9 in rainbow trout, or the turquoise killifish, where a new member of the TGF β factors is doing the job). *Foxl2* is admittedly an important female promoting sex determination gene, but so far did not show up as primary sex determiner. Second, this way of proposing candidates works only for allelic diversification, but not for another major mechanism, how new sex determining genes evolve, gene duplication and translocation. It suffices to inform the reader that the *Rana* proto-chromosomes that most often evolved to sex chromosomes have these candidate genes and more work is needed. Why they are good candidates is told in the references and if the reader is interested he/she will find all this information there or in reviews. The first paper that pointed into the direction of the hypothesis that “some chromosomes are better at sex” should be cited as well.
- Lines 447 – 451: Again this point is not relevant and just speculation. The absence of evidence that in other species those genes have jumped does not mean that they cannot do it in the Ranidae. The salmonid sex determining gene is a duplicate of a regular immune gene for which also no evidence exists that it is a “jumping” gene. On the other hand transposon can capture regular genes or trans-acting factors or two transposons flanking a gene can mobilize it. Transposons that harbour e.g. testis specific enhancer sequences could jump around and depending on preferred landing sites activate a gene on Chr01, 03 or 05. Such discussions and hypotheses might, however, be better suited for a review rather than a regular research paper.
- Lines 915 ff: It needs to be explained why *R. japonica* and *R. pipiens* have two colours and what the arrows following the species name indicate (as opposed to the arrows that on the branches). What is indicated by the question mark (even if one could guess that it means “unknown”)?
- Lines 932 – 933: The legend is insufficient. Much more explanation is needed to evaluate what is shown here, eg. what the landscape left of the chromosome reference indicates, the bars mean (that I guess give the gene density), or what the height of the columns for each species indicates (RAD-tags per how many cM or kb?).

Dear reviewers,

We sincerely thank you for your comments on our manuscript.

The responses below hopefully address your concerns adequately. We give line numbers in our responses, which refer to the versions of the revised manuscript and supplementary text that contain track changes – hopefully this makes it easier for you to see the changes we have made. These versions have “TC” in the file name. Please note that the references added in these versions are not in the reference lists for these versions, but are included in the bibliographies of the versions with “revised” in the file name, in which all track changes have been accepted.

Reviewer #1 (Remarks to the Author):

Jeffries et al identify male and female heterogametic sex chromosomes across multiple frog species, with the aim to characterize the rate of turnover between XY and ZW chromosomes and establish whether some autosomes are predisposed to be sex-linked. Identifying the factors driving sex chromosome turnover is a hot topic in evolutionary biology yet identifying sex chromosomes in groups with high rates of turnover is not a trivial task. The manuscript is extremely well written and I applaud the authors for their clear and transparent methods. This is the first paper to my knowledge to attempt to rigorously distinguish between difference scenarios promoting sex chromosome turnover and will likely be of broad interest to many people. Please see my detailed comments below.

1. For the ancestral state reconstruction, the authors only consider 5 possible states. Therefore, for species where sex-determination is unknown, only 5 possible chromosomes were permitted to be considered for sex-linkage. This seems rather strange to me and might instead bias the analysis to find that the same chromosomes are more often recruited for sex-linkage than in reality. It would be more appropriate not to impose any assumptions on the data at this point. Indeed, from Fig S6, although it is not significant, it seems there might be evidence that Chr 7 is sex-linked in *R. pipiens*. This would then be 6 possible states. The authors should redo this allowing

all chromosomes to be potentially sex-linked. Given this analysis forms the basis for the central question of the study, rerunning it without a priori assumptions is extremely important. Are the results comparable?

R: Our decision to use only 5 states was an attempt to avoid over-parametrisation, reduce noise, and increase power in the ancestral state reconstruction. It was made under the assumption that the stochastic mapping could not favor a transition to a state that was never used by any of the sampled species. That said, the reviewer is correct that this is the central analysis of the study, and so we should formally test for any effect of this assumption. To that end, we re-ran the stochastic mapping analyses using a state probability matrix that gives equal probability to all 10 possible chromosome states at the unknown tips. The results are attached below in Fig. R1, which has the same format as Fig. 1 of the manuscript for comparison. It is hopefully obvious from the node pie charts in this figure, that the relative likelihoods of states throughout the tree are very similar between the 5-states and 10-states analyses. There is only one noticeable difference between the results of the two analyses, that is, transition 6 has moved from the branch to (*R. kukunoris*, *R. chensinensis*) to the branch to (*R. ornativentris*, (*R. kukunoris*, *R. chensinensis*)) as shown by the red arrow on Fig. R1. However this does not change the number or the biased pattern of the turnovers. We therefore hope that this satisfies the reviewer that the observed recruitment bias in our data is not an artifact of this analysis. We still feel that our initial analysis is more suited to inclusion in the manuscript as it is less “noisy” (and especially makes Fig. S8 much easier to interpret), but we have included a couple of lines (731-732) in the methods to mention that we have formally checked this assumption.

2. In the discussion, the authors mention that the pattern of sex chromosome turnover is consistent with a mutation load or drift hypothesis. The majority of the discussion (and abstract) is then focused on the mutation load hypothesis. The justification seems to be that male heterochiasmy would promote rapid accumulation of deleterious mutations. However, occasional recombination has been shown in this group and surely this would purge the deleterious mutations? There needs to be more focus on the impact of this in the discussion. To me, it makes the mutation load hypothesis

rather unlikely. What would be mutation and recombination rates need to be in order to for this scenario to be plausible?

R: XY recombination does indeed occur in some populations of frogs, in association with rare events of sex reversal ('leaky GSD'): as meiotic recombination depends on phenotypic sex, X and Y occasionally recombine in populations where sex-reversed XY females occur, preventing X-Y differentiation. There are other populations, however, that display strict GSD: XY individuals always develop as males, so that X and Y never recombine and are therefore well differentiated. These are the populations in which we expect deleterious mutations to accumulate on the Y, ending up with a sex-chromosome turnover. This polymorphism in XY differentiation among conspecific populations seems a widespread feature in frogs; leaky GSD might thus be responsible for our failure to identify sex chromosomes in several of our samples. It is however clear from our previous work on *Rana temporaria*, as well as from the present data on other Ranidae (Fig. 3), that XY differentiation does occur in many frog populations, in association with strict GSD. We added the following sentence in the Discussion (lines 443-452) to clarify this point:

“As meiotic recombination depends on phenotypic sex (not on genotypic sex), X and Y chromosomes may still recombine occasionally in the rare sex-reversed XY females that occur when alleles at the sex locus show incomplete penetrance (Rodrigues et al. 2018), preventing X-Y differentiation. This possibly accounts for our inability to identify sex-linked markers in some of our samples. However recombination will immediately cease along almost the entire length of the X and Y chromosomes as soon as full-penetrance alleles are fixed, which is clearly the case in some *R. temporaria* populations (Rodrigues et al 2014, 2015, 2016) as well as in populations of other Ranid species that display extensive X-Y differentiation (Fig. 3).”

Furthermore, we also note that the only exception to the general pattern of maintenance of male heterogamety (in *G. rugosa*) has been explicitly assigned to sex-ratio selection. Hence this leaves 11 turnovers to be explained, all of which keep male heterogamety. We think this further increases the likelihood of the hot-potato model over the genetic drift model (l. 427-429).

3. Given the importance of correctly identifying sex chromosomes for the conclusions of this paper, the authors do a fantastic job in investigating inconclusive patterns of sex linkage. In Fig S2 they show that the signal of XY and ZW sex-linked markers on the same chromosome is due to paternal X-specific SNPs. However, they show this only for *R. montezumae* (panel e). Can they also show the same is true for *R. italica*, *R. chrichahuensis* & *R. kukunoris*? Or are there any independent lines of evidence supporting an XY system in *R. italica*, *R. chrichahuensis* & *R. kukunoris* as in *R. arvalis* where there is PCR validation? Given the uncertainty over these species, it would be reasonable to conduct additional PCR validation tests.

R: We would very much like to test this theory in these species, unfortunately in some cases we are constrained by the samples that we have available. However, having put more thought and analyses into it, we have now extended our arguments for XY systems in these species. As this section is now not trivial, we have moved it to the supplementary text, in order to give us the space to address it in sufficient detail. Please see section S.1 of the supplementary text for full arguments. We have altered the main text to briefly mention this topic and cite the supplementary text (lines 209 – 220). We also summarise these arguments briefly below.

False ZW signal can occur in two ways from X-specific markers. The first, already described in Fig. S2, relates to paternal-X specific alleles, and only applies to datasets from families. The second is from null alleles on the Y chromosome, meaning that males will be hemizygous for those RADtags. This will cause false signal of homozygosity in the data and potentially of ZW sex-linkage. The latter was not explicitly mentioned in this section previously, we now describe it in the supplementary text section and Fig. S2 alongside the paternal-X theory.

The only way to test the paternal-X specific SNP theory in the family data is to examine the genotypes in the parents as we have done in *R. montezumae*, where we were lucky to have the mother of one family. Unfortunately no parental samples were available for *R. italica*. We cannot see how PCRs of sex linked markers could help us in this situation, as only parental genotypes would allow us to test this hypothesis (but we would be happy to hear how). That said, for *R. italica*, there is indeed an

independent line of evidence in support of an XY system. When we align the ZW-like markers to the reference, we see that they are distributed along the entire length of the chromosome, which is consistent with XY sex-linkage in frogs due to the strong heterochiasmy of the system. However if the system were in fact ZW, we would expect to see a much more conventional pattern of sex-linkage – i.e. clustering of sex linked markers around the sex determiner, due to the high rate of recombination in females - we expand on this in Supp. Text. S1.

For *R. chricahuensis*, on close examination, we realised that there was an error in denoting the number of loci found using the real sex assignments in the randomisation scripts (it was coded as 3,3,2 ZW markers for methods 1,2 and 3 respectively, whereas the true number of ZW-like loci identified was 2,0,1. These numbers are in fact not significantly outside of the null distribution from the sex-assignment randomisations. We apologise for this error, have removed discussion of *R. chricahuensis* from this section of the manuscript, and updated Fig. S1.

For *R. kukunoris*, these samples come from population data. We therefore expect the ZW signal to come from X-specific RADtags (i.e. Y null alleles). We can test for this by comparing male vs female coverage at ZW-like markers. If indeed the males are hemizygous, these loci should have lower coverage in males than in females. For *kukunoris* this test was positive, males showed a significant reduction in coverage relative to females at these loci. Thus we are confident that these are X-specific RADtags.

Taken together, we feel we can make a strong case for XY heterogamety in each of the four species where XY and ZW markers pass validation.

4. The evidence in support of the expected patterns of heterogamety under drift are detailed in a paper that is not publicly available. The authors should give further detail here to explain this hypothesis.

R: We provide now a few more details on the results of these individual-based simulations (lines 141-144), but this paper is now in fact published and fully cited (Saunders, P. A., Neuenschwander, S., and Perrin, N. (2018). Sex chromosome

turnovers and genetic drift: A simulation study. *J. Evol. Biol.* doi:10.1111/jeb.13336.)

5. Fig S3 panel c). Checking the effect of sample size on false positive rate is great. However, this would be more appropriate if it was conducted on the Y1 subsample, so the starting point is significant evidence of XY. I am not sure in its current form that the figure supports the author's conclusion that the tests can confidently identify sex-linked markers down to sample size of 7M and 7F because 1) the ZW markers are false positive, 2) everything is significant at all sample sizes. A similar figure testing the effect of unbalanced male and female sampling for the identification of ZW and XY markers would also be beneficial here.

R: It was not explicitly stated, but the downsampling analyses were indeed conducted only using Y1 males – apologies, this is now made more clear on the figure. Regarding the second point: we can be confident that many of the markers identified are sex linked down to 7M, 7F, however the reviewer is right that the increase in the number of false positives as the sample size decreases does indeed lower the confidence that any given marker is truly sex linked. We have edited / added to (what is now) Supplementary text S3 (lines 154-167 of Supplementary text) and the figure legend to reflect this and to state that false positive rate increases as sample size decreases, and this means that validation of these markers is imperative in such cases. We have also added two more downsampling analyses, one skewed towards males and the other towards females. Fig S3 has been updated to include these and they are also referred to in the Supp. Text. S3.

L136 Missing 'more' likely

R: Added

L493 Can the authors provide extra explanation for why these parameters will ensure over-merged loci resulting from repetitive elements are removed. They seem rather arbitrary at the moment.

R: It was perhaps not clear that we were referring only to the `-max_obs_het` filter in the context of filtering for overmerged loci. We have made this clearer and added a

few extra sentences on the use of this parameter (l. 566-574).

L504 Font changes in this paragraph

R: Fixed

Fig S6 What is the significance of the dotted box around Chr 2 in *R. latastei*? Why does *P. perezi* not have one?

R: This was an artifact of a previous version of the figure, its purpose was just to highlight borderline examples. We have now removed it.

Fig S8 What is the significance of the * next to the species name? This is not explained in the figure legend.

R: Again an artifact of a previous version which we have now removed. Thank you for pointing these out.

Reviewer #2 (Remarks to the Author):

The authors characterize the sex chromosomes of many ranidae species, and find a very high rate of turnover, potentially explaining their homomorphy despite the lack of recombination in males. I think that these are very interesting and well written results based on a sound analysis, but have comments about the interpretation of some of the patterns. I also feel that previously reported findings could be made clearer throughout the text.

1. Maintenance of male-heterogamety and the hot potato model:

The authors view the maintenance of male heterogamety as evidence for the hot potato model. However, this is also expected if the same gene is determining sex at different locations, and if the same genes are recruited convergently for sex determination.

Although I agree that the lack of recombination in males makes ranidae likely to suffer from a quick accumulation of deleterious mutations on the Y, the authors don't

have any evidence of it happening in the species under study. I think that the discussion of these patterns needs to be a bit more balanced, and other models that predict the maintenance of male-heterogamety mentioned.

R: The reviewer is perfectly correct that we have no direct evidence yet for the predicted accumulation of deleterious mutations on the non-recombining Y chromosomes. We mention this in the Discussion, pointing out that the quantification of deleterious mutations on the Y chromosomes of frogs would constitute a welcome empirical extension of the present work (l.463). However, our data (Fig. 3) do show that many sex-linked loci have fixed different alleles on the X and the Y chromosome, which necessarily implies that these chromosomes have stopped recombining for a significant amount of time. Accumulation of deleterious mutations is the unavoidable consequence of recombination arrest, and should be further amplified by the fact that the largest part of the sex chromosome stops suddenly to recombine, generating strong Hill-Robertson interferences at a large scale (l. 452-455).

Regarding alternative hypotheses: we had already acknowledged in the previous version (l. 505) that we cannot rule out the possibility of a translocation of the sex-determining gene (while noting however that the candidate SD gene in *R.temporaria* is autosomal in *G. rugosa*, and vice versa). We do not see convergent recruitment as an alternative to the hot potato: the driver (ultimate cause) can be the accumulation of deleterious mutations, and the same genes recruited convergently (proximate cause). This is actually what we think is happening. But convergence to the same gene does not impose the maintenance of heterogamety: mutation of any gene in the male cascade can be either masculinizing or feminizing. A mutation of *Dmrt1*, for instance, will be masculinizing if it upregulates its expression, and feminizing if it downregulates its expression. This is now clarified in the Discussion (lines 514-521)

2. Novelty of the patterns described:

The same group had previously characterized the sex chromosomes of *R. arvalis* (XY, chr1), yet this species seems to count towards the 20 frog species for which "the sex determination system and identity of the sex chromosome were previously unknown". Similarly, *R. berlanderi*, *R. pippiens*, *R. sphenoccephala*, *R. japonica* were at least partly characterized in the literature (and in several cases the sex chromosome

assignment shown in figure 1 came from the literature and not from the new data, which did not have enough power to detect the SD chromosome).

Of the 7/20 species for which the sex chromosome could be determined, 2 were confirmations.

This means that at least 7 of the transitions were reported from the literature.

I think it should be clearer in the main article text and figures what is new data and what is previously reported.

R: The reviewer is right, this was not clear. We have now added/amended sentences throughout the main text to highlight which species and which information comes from our data, and which comes from the literature. See lines 537-538, 639-641 of the methods, and lines 192-194, 292-295 of the results. We did originally have a version of Fig. 1 showing the information for the origin of the information, but as this figure already has a lot of information on it, we opted to remove this as it became too complicated. We hope that with the amendments in the text, along with tables S1 and S3 which show this information explicitly, this is now clear.

3. Pelophylax and Glandirana have different names in figure 1 and table s3.

R: Amended, thank you.

Reviewer #3 (Remarks to the Author):

The manuscript reports on the rate of sex chromosome turnover among frogs. For this study RADtags were produced from 20 species of the frog family Ranidae to add additional data for species where the sex chromosome system in this family was not already known. In total, except for four species they obtain information for the 28 species that they selected. Plotting the sex chromosome data on the phylogenetic tree revealed the highest rate of sex chromosome turnover in vertebrates known so far. Their data are discussed in the framework of major hypotheses of sex chromosome evolution and give important insights into the evolutionary forces that shape this process. The paper is well written and is of outstanding importance for the research field. It should be of interest also for scientists that work in related areas. In general, it

is very well suited for publication in Nature Communications. The experimental and bioinformatics analysis were all done with state of the art technology and are described in sufficient detail. The important drawback of the unavailability of a chromosome-anchored high quality reference genome for this family of frog species is elegantly circumnavigated with clever usage of trans-mapping approaches.

There are, however, several points that need to be considered, before this paper can be published:

- Lines 198 ff: The lack of information from the RAD-tags in some species raises concern. It is pretty likely that for *R. macrocnemis*, *chinensis*, *ornaventr* and *uenoi* not enough individuals were analysed, probably due to unavailability. Gamble et al. may just have been lucky with low numbers of individuals or the enzyme used. In other studies the experience is that increasing the sample size improves the outcome of RAD-tag analyses. Another possibility is that these 4 species have either molecularly undifferentiated sex chromosome (remember Fugu!) or have polyfactorial genetic or environmental sex determination. Anyway, it is difficult to understand why these species were then included in the overall analysis, in particular in Figure 1, which gives the main message. There is no mention of this issue in the whole paper and I would see it as improvement to delete them.

R: Our failure to find sex-linked markers in some samples was already shortly discussed (l. 247-253). We indeed consider likely that these species/populations have no differentiated sex chromosomes, either because the SDR is very small, or sex determination is not genetic. Both situations have been actually documented in *Rana temporaria*, where sex determination ranges from strictly genetic to entirely non-genetic depending on populations (e.g. Rodrigues et al 2014, 2015, 2016, Brelsford et al. 2016). A similar polymorphism in sex-chromosome differentiation also occurs in other Ranidae (e.g. *R. iberica*), and is likely a widespread feature across Ranidae. We now expand more on the mechanisms underlying this polymorphism in sex-chromosome differentiation, and mention this as a likely cause for our inability to identify sex-linked markers in some samples (lines 443-453). However, we think important to also report these negative results in our paper because they might be biologically meaningful, and thus prefer not to drop reference to these species.

- Line 293: *G. rugosa*, not *R. rugosa*

R: Fixed, thank you.

- Lines 335 – 362: In this context it is of interest to compare the situation in the Ranidae not only to the genus *Oryzias*, but also to Salmonids and the Geckos. Both groups are also well-known for high sex chromosome turnover and the reader can expect to be told how these compare in terms of frequency and time with the frogs. In particular, the finding of 19 transitions in the geckos compared to the 13 in Ranidae needs discussion.

R: We completely agree that it would be nice to compare our rate of transition to that of other systems. Unfortunately this is not easy to do in a formal way as it would require both reliable information on the chromosome pair that determines sex for a wide array of species within a clade, and a robust dated phylogeny of these species. Frequent turnover of sex determination systems or sex chromosomes have been described in many taxa. Indeed, as the reviewer mentions, Gamble et al. (2015) have described a high rate of transitions from temperature to genetic sex determination in geckos, but comparing sex chromosome turnovers with sex determination system transitions is not informative. Furthermore the 19 transitions described in Gamble et al. (2015) is not the parsimonious number in this data set (closer to 8). In Salmonids, several different sex chromosomes are also known, suggesting frequent turnovers (reviewed in Sutherland et al 2017). However, no study has attempted to summarise these findings and to place them in a phylogenetic framework so far. Finally, several sex chromosome turnovers have indeed been described in *Oryzias* fishes. Myosho et al. (2015) have documented the sex chromosome in several species and summarised the available literature. They highlighted some turnovers, but their analysis is not exhaustive and they did not conduct an ancestral states reconstruction analysis, making it impossible to estimate on which branches turnovers occurred. Additionally, the age of the *Oryzias* radiation is not yet settled, with a threefold difference between studies (see Setiamarga et al. 2009 and Mokodongan & Yamahira 2015) which would vastly influence turnover rate estimates.

However, as we feel that this comparison is an important one, and will become more important as similar studies are published in coming years, we now propose a method to calculate an approximate rate of transition across our tree, which can be performed on other such studies. We hope that future studies will use this method, or one similar to perform this comparison.

In the mean time, we believe we can make a strong argument based on anecdotal evidence, that ours is the fastest rate of sex chromosome turnover reported to date as we are one of very few studies to have performed an ancestral state reconstruction and thus truly estimate the number of transitions. We have added to this anecdotal argument in the main text on lines 389 – 400.

- Lines 386 – 403: The authors use what they call “extreme patterns” of heterochiasmy as argument against the importance of sexually antagonistic genes in the process of sex chromosome evolution and as support for their “hot potato” model. Here the reader will expect a comparison of the extent of heterochiasmy in the Ranidae with the rates known for most mammals and fish, e.g. zebrafish. According to the Haldane-Huxley rule achiasmy occurs in the heterogametic sex. Is there an inversion of the pattern in the W/Z *G. rugosa*?

R: We have added some comparisons of heterochiasmy with other species on lines 435-439. The Haldane-Huxley rule holds for achiasmatic, but not heterochiasmatic species. And in fact we know specifically in *G. rugosa* that the system of heterochiasmy has not switched with the system of heterogamety (i.e. females still recombine much more than males). We have actually added a section in the supplementary text in response to another reviewer comment which addresses this issue in more detail - Please see lines 110 – 126.

- Lines 417 – 439: This part should be shortened considerably, because it is highly speculative. The given examples of candidate genes on the most frequently used chromosomes are clearly very suggestive, but considering that hundreds of genes will be located in the regions identified by the RAD-tags, many other candidate genes will exist. First, not always the usual suspects can be made responsible (see IGF9 in rainbow trout, or the turquoise killifish, where a new member of the TGF β factors is doing the

job). Foxl2 is admittedly an important female promoting sex determination gene, but so far did not show up as primary sex determiner. Second, this way of proposing candidates works only for allelic diversification, but not for another major mechanism, how new sex determining genes evolve, gene duplication and translocation. It suffices to inform the reader that the Rana proto-chromosomes that most often evolved to sex chromosomes have these candidate genes and more work is needed. Why they are good candidates is told in the references and if the reader is interested he/she will find all this information there or in reviews.

The first paper that pointed into the direction of the hypothesis that “some chromosomes are better at sex” should be cited as well.

R: We acknowledge that this section is speculative, but we also do not see the harm in this speculation in the discussion as long as it is advertised as such. It seems to us, very appropriate here to point out the locations of several of the usual suspects – Dmrt1 and Sox3 for example, but also Foxl2. Not only is Foxl2, to our knowledge, the only gene with a prominent role in sex determination on Chr05, but its direct interaction with Dmrt1 – the likely ancestral sex determiner for the family – makes it, in our opinion, more than worthy of mention. Thus, we would prefer to keep this section in.

- Lines 447 – 451: Again this point is not relevant and just speculation. The absence of evidence that in other species those genes have jumped does not mean that they cannot do it in the Ranidae. The salmonid sex determining gene is a duplicate of a regular immune gene for which also no evidence exists that it is a “jumping” gene. On the other hand transposon can capture regular genes or trans-acting factors or two transposons flanking a gene can mobilize it. Transposons that harbour e.g. testis specific enhancer sequences could jump around and depending on preferred landing sites active a gene on Chr01, 03 or 05. Such discussions and hypotheses might, however, be better suited for a review rather than a regular research paper.

R: We think there is a slight misunderstanding here. We are not claiming that Dmrt1 or Sox3 cannot associate with transposons, but only that the candidate sex-determining gene in *G.rugosa* (Sox3) is autosomal in *R. temporaria*, and reciprocally the candidate SD gene in *temporaria* (Dmrt1) is autosomal in *rugosa*. Hence there is

no support for the idea that sex is determined by the same gene on different chromosomes in this specific case. We have reformulated the sentence to make this more clear (l. 513).

- Lines 915 ff: It needs to be explained why *R. japonica* and *R. pipiens* have two colours and what the arrows following the species name indicate (as opposed to the arrows that on the branches). What is indicated by the question mark (even if one could guess that it means “unknown”)?

R: Added clarification to Figure 1 legend.

- Lines 932 – 933: The legend is insufficient. Much more explanation is needed to evaluate what is shown here, eg. what the landscape left of the chromosome reference indicates, the bars mean (that I guess give the gene density), or what the height of the columns for each species indicates (RAD-tags per how many cM or kb?).

R: Added clarification to Figure 3 legend.

Figure R1. Sex chromosome turnovers across 28 true frog species inferred from ancestral states reconstruction using 10 possible states. Sex-determination system and sex-chromosome identities come from both RADseq (Table S1) and literature (Table S2) data. Karyotype (top left, chromosomes not to scale) shows number of species using each chromosome for sex determination. Coloured arrows show the branch on which inferred turnovers occur based on the stochastic mapping analyses (Fig. S8) and the pie charts at nodes represent the proportion of simulated trees in stochastic mapping with each of the states at that node. Grey in karyotype and node pies denotes chromosomes which have never been observed to be used as sex chromosomes in this system. Empty (white) circles at tips represent unknown sex-chromosome identities in both tips and turnover arrows 1 & 2. As there was not enough high quality sequence information for *P. porosus*, it could not be included in the phylogenetic reconstruction or stochastic mapping. Its position here is inferred from that in ⁴⁴.

REVIEWERS' COMMENTS:

Reviewer #1 (Remarks to the Author):

The authors have done a very satisfactory job in addressing the majority of my concerns. I have only minor comments regarding these revisions.

Dryad number is missing

L193 Extra gap between 'exhibit' and 'the'

However, I still have concerns about the point I raised regarding the mutation-load selection hypothesis. The authors argue this is the driver of sex chromosome turnover across this group due to the preservation of XY systems. However, surely this is biased by the absence of male recombination across these species, and this in itself would bias the formation of new sex chromosome systems to be XY? For example, if sexual antagonism were driving sex chromosome turnover, the system would also be more likely to be XY just because of the absence of recombination in males. It is interesting that a pattern of more XY systems is also observed across Diptera, which also have male achiasmy (Vicoso & Bachtrog 2015, PLoS Biology). Whilst the predictions in the introduction seem reasonable for species with recombination in both sexes they need to be reconsidered for situations with male achiasmy. This alternative explanation should be discussed in the manuscript. The manuscript is a great piece of work and is very suitable for publication in Nature Communications once these considerations are incorporated into the manuscript.

Reviewer #2 (Remarks to the Author):

The results presented here are exciting, and the updated manuscript is in my opinion suitable for publication. Regarding my previous comments:

1. I still think that the strength of the evidence for the hot potato model is a little bit overstated, e. g.:

abstract: implying a key role for mutation-load

104 accumulation in non-recombining genomic regions  "suggesting"?

L. 196 "suggesting that mutation-load selection is

197 the predominant driving force in this system."  "an important driving force"?

I am also not sure that the HPM and other selective processes are exclusive, as suggested in the text. For instance, you say that:

"This bias runs against expectations from models where either

413 sex-ratio selection or sex-antagonistic selection drive turnovers^{31–33,50}."

But if sexual antagonism is driving turnover, AND mutationally loaded Ys cannot be fixed, you would preserve heterogamety as well (but the causal force of turnover would not be the HPM).

2. I still find this sentence slightly confusing:

"Here we use Restriction site Associated DNA sequencing
190 (RADseq) to search for sex-determination system and the identity of the sex
chromosomes in
191 19 frog species for which they were previously unknown. "

Since 4/20 species are listed in "Supplementary Table 3. Results of literature search for sex determination system and sex chromosome identity for Ranid species."

I would also perhaps mention that the analysis was not successful for all, i.e.:

" Of the 12 species for which sex-linked
260 markers were confidently identified, 7 species passed these alignment validation tests
261 (Supplementary Fig. 5, Supplementary Table 1). "

These are both very minor points though, and I leave it up to the authors to decide whether to rephrase.

Reviewer #3 (Remarks to the Author):

none

Responses to reviewers comments

We sincerely thank the reviewers once again for their comments. Please find point by point responses below.

Reviewer #1

The authors have done a very satisfactory job in addressing the majority of my concerns. I have only minor comments regarding these revisions.

Dryad number is missing

R. Changed submission to Figshare, doi now included.

L193 Extra gap between 'exhibit' and 'the'

R. Ammended

However, I still have concerns about the point I raised regarding the mutation-load selection hypothesis. The authors argue this is the driver of sex chromosome turnover across this group due to the preservation of XY systems. However, surely this is biased by the absence of male recombination across these species, and this in itself would bias the formation of new sex chromosome systems to be XY? For example, if sexual antagonism were driving sex chromosome turnover, the system would also be more likely to be XY just because of the absence of recombination in males. It is interesting that a pattern of more XY systems is also observed across Diptera, which also have male achiasmy (Vicoso & Bachtrog 2015, PLoS Biology). Whilst the predictions in the introduction seem reasonable for species with recombination in both sexes they need to be reconsidered for situations with male achiasmy. This alternative explanation should be discussed in the manuscript.

R: The reviewer is right in principle. We have now incorporated the caveat that SA selection might indeed bias turnovers towards XY systems, given the low rate of male recombination (l. 369-382). However, we also make the point that SA selection is not expected to generate continuous cycles of turnovers, as documented here (Blaser et al 2013, 2014). Even though an autosomal male-beneficial mutation might indeed trigger an initial XY-to-XY transition, it should strongly oppose any further transition once sex linked. Furthermore, genomic investigations do not support a role for SA genes in sex-chromosome evolution in *Rana temporaria* (Ma et al 2018 Genes). Overall, we still think that data in hand seem more compatible with a role for mutation load.

The manuscript is a great piece of work and is very suitable for publication in Nature Communications once these considerations are incorporated into the manuscript.

Reviewer #2

The results presented here are exciting, and the updated manuscript is in my opinion suitable for publication. Regarding my previous comments:

1. I still think that the strength of the evidence for the hot potato model is a little bit overstated, e. g.:

abstract: implying a key role for mutation-load accumulation in non-recombining genomic regions  "suggesting"?

R: changed as suggested

L. 196 "suggesting that mutation-load selection is the predominant driving force in this system."  "an important driving force"?

R: changed as suggested

I am also not sure that the HPM and other selective processes are exclusive, as suggested in the text. For instance, you say that:

"This bias runs against expectations from models where either
413 sex-ratio selection or sex-antagonistic selection drive turnovers^{31–33,50}."

But if sexual antagonism is driving turnover, AND mutationally loaded Ys cannot be fixed, you would preserve heterogamety as well (but the causal force of turnover would not be the HPM).

R: The reviewer is correct in principle, and we added this comment (l. 369-382). However, we also make the point that SA selection is not expected to generate continuous cycles of turnovers, as documented here (Blaser et al 2013, 2014).

2. I still find this sentence slightly confusing:

"Here we use Restriction site Associated DNA sequencing
190 (RADseq) to search for sex-determination system and the identity of the sex
chromosomes in
191 19 frog species for which they were previously unknown. "

Since 4/20 species are listed in "Supplementary Table 3. Results of literature search for sex determination system and sex chromosome identity for Ranid species."

R. deleted "for which they were previously unknown". And added an ammendment to the next sentence just to state the total number of species used. This gets away from stating here the complicated overlaps between the new and existing data.

I would also perhaps mention that the analysis was not successful for all, i.e.:

" Of the 12 species for which sex-linked
260 markers were confidently identified, 7 species passed these alignment validation
tests

261 (Supplementary Fig. 5, Supplementary Table 1). "

R. We say this already here:

173 Of the 20 species for which we analysed RADseq data, 12 had sex-linked marker sets that passed these sex-permutation validation tests for at least one method

These are both very minor points though, and I leave it up to the authors to decide whether to rephrase.